# Effect of Collagen Matrix on Doxorubicin Distribution and Cancer Cells’ Response to Treatment in 3D Tumor Model

**DOI:** 10.3390/cancers14225487

**Published:** 2022-11-08

**Authors:** Irina Druzhkova, Elena Nikonova, Nadezhda Ignatova, Irina Koryakina, Mikhail Zyuzin, Artem Mozherov, Dmitriy Kozlov, Dmitry Krylov, Daria Kuznetsova, Uliyana Lisitsa, Vladislav Shcheslavskiy, Evgeny A. Shirshin, Elena Zagaynova, Marina Shirmanova

**Affiliations:** 1Research Institute of Experimental Oncology and Biotechnology, Privolzhsky Research Medical University, 603005 Nizhny Novgorod, Russia; 2Lomonosov Moscow State University, 119991 Moscow, Russia; 3Sechenov First Moscow State Medical University of the Ministry of Health of the Russian Federation (Sechenov University), 119991 Moscow, Russia; 4School of Physics and Engineering, ITMO University, 9 Lomonosova St., 191002 St. Petersburg, Russia; 5Lobachevsky State University of Nizhny Novgorod, 603022 Nizhny Novgorod, Russia

**Keywords:** drug delivery, cancer metabolism, NAD(P)H, extracellular matrix, Doxorubicin, microfluidic chip, glycolysis

## Abstract

**Simple Summary:**

Understanding of the interactions of chemotherapeutic agents with cancer cells in the presence of extracellular matrix (ECM) is necessary for improving drug delivery and treatment efficacy. In this study, using 3D collagen-based models, microfluidic devices, and multiphoton fluorescence microscopy, we visualized delivery of doxorubicin in ECM and its effects on cellular metabolism. We found that collagen, especially unstructured, plays a protective role for cancer cells because it impedes the delivery of the drug and changes cellular metabolic status to a more glycolytic status. The direct effect of the drug on mitochondria also contributes to its higher cytotoxicity in the absence of collagen. These results suggest that modification of ECM structure can be a potential strategy for improving efficacy of chemotherapy with doxorubicin.

**Abstract:**

The extracellular matrix (ECM) plays an important role in regulation of many aspects of tumor growth and response to therapies. However, the specifics of the interaction of chemotherapeutic agents with cancer cells in the presence of collagen, the major component of ECM, is still poorly investigated. In this study, we explored distribution of doxorubicin (DOX) and its effects on cancer cells’ metabolism in the presence of collagen with different structures in 3D models. For this, a combination of second harmonic generation imaging of collagen and multiphoton fluorescence microscopy of DOX, and metabolic cofactor NAD(P)H was used. It was found that collagen slowed down the diffusion of DOX and thus decreased the cellular drug uptake. Besides nuclei, DOX also targeted mitochondria leading to inhibition of oxidative phosphorylation, which was more pronounced in the cells growing in the absence of collagen. As a result, the cells in collagen displayed better viability upon treatment with DOX. Taken together, our data illustrate that tumor collagen contributes to heterogeneous and sub-optimal response to DOX and highlight the challenges in improving drug delivery and efficacy.

## 1. Introduction

Most studies of tumor resistance to anticancer drugs consider only its cellular or genetic causes [1]. However, a complex tumor microenvironment has been demonstrated to influence drug sensitivity [2]. In the microenvironment, the extracellular matrix (ECM), which represents the complex network of macromolecules surrounding cells, is of special interest. The main components of the ECM, such as collagens, proteoglycans, and glycoproteins, provide not only a passive mechanical support for tumor cells, but actively regulate their functionality [2,3]. Different ECM components that interact with cell surface receptors are able to mediate cell adhesion and signaling, and thereby regulate different processes such as proliferation, differentiation, migration, and apoptosis [2,4]. In addition, the ECM strongly affects transport mechanisms, metabolism, oxygenation, and immunogenicity of solid tumors, and thus, along with biological behavior of tumor cells, it also regulates their response to therapy [3,4].

Many anticancer drugs have limited distribution from blood vessels in solid tumors by both diffusion and convection because the ECM serves as a barrier for their penetration, which restricts their effectiveness [5]. It is obvious that if anticancer drugs are unable to access all cells within a tumor, their effectiveness will be compromised since residual cells retain the ability for disease relapse [1]. Notably, not only the amount of ECM molecules is important, but also their composition and organization within a tumor tissue. The approaches targeting the ECM structure to improve the efficacy of chemotherapy are now well developed. Numerous studies have demonstrated that, for example, the inhibition of the ECM maturation process or application of ECM-disrupting enzymes such as hyaluronidase and collagenase result in better penetration of chemotherapeutic drugs to tumor cells in experimental models [6,7,8,9]. Meanwhile, the specifics of the interaction of chemotherapeutic agents with cancer cells in the presence of ECM components and the role of the ECM in the response of cancer cells to chemotherapy are not fully understood. 

Doxorubicin (DOX) is a chemotherapeutic agent of the anthracycline group, routinely used for treatment of soft tissue and bone sarcomas, breast, ovary, bladder, thyroid, and some other cancer types [10]. Two main mechanisms of action of this drug on cancer cells have been established: (i) intercalation into nuclear DNA with the formation of DOX-DNA adducts and inhibition of topoisomerase II leading to disruption of DNA repair, and (ii) generation of reactive oxygen species and hydrogen peroxide by semiquinone radical causing oxidative stress [11,12]. Recent studies suggest that DOX interferes with the mitochondrial function, and these alterations are proven to determine its cardiotoxicity, but are still poorly investigated in cancer. At the same time, an extensive reprogramming of metabolism in cancer cells under DOX treatment favors an acquisition of the drug resistance [13,14]. 

Multiphoton laser scanning microscopy (LSM), with the option of second harmonic generation (SHG) imaging and fluorescence lifetime imaging (FLIM), has become a powerful instrument in preclinical studies. In drug research, fluorescence intensity imaging is usually used to follow an accumulation and subcellular distribution of a drug, while FLIM reports mainly on the drug–target interactions. Another area of FLIM applications is metabolic imaging using autofluorescence of the cofactors, the reduced form of nicotinamide adenine dinucleotide NAD(P)H and oxidized flavins FAD/FMN, providing information about the relative activity of bioenergetic pathways [15,16]. Of these two types of cofactors, NAD(P)H fluorescence parameters are easier to interpret, and therefore its fluorescence is often used as a metabolic indicator. If a drug and cofactor fluorescence are spectrally separated, monitoring of the drug state and cellular metabolism can be performed simultaneously in the same individual cells with high sensitivity and molecular specificity. In turn, SHG microscopy allows the assessment of architectural features of collagen, which can be used to evaluate the effects of ECM on the drug response. Previous LSM/FLIM studies of DOX, which fluoresces in the yellow spectral range (ex. max. 490 nm, em. max. 595 nm), have provided the insight into the localization and microenvironment of the drug within tumor cells and tissues [17,18]. Using FLIM, NAD(P)H metabolic shifts induced by DOX in cultured cancer cells have been identified in separate studies [19,20]. However, in most of the investigations, the ECM context is missing.

The purpose of our study was to analyze the distribution of DOX in cancer cells and its effects on cancer cells’ metabolism in the presence of collagen in a 3D model, using a combination of SHG imaging of collagen, fluorescence intensity, and lifetime imaging of DOX and NAD(P)H. To mimic drug delivery to tumor cells, a microfluidic system was developed. In a 3D collagen model, cancer cells were loaded alone or in the co-culture with normal fibroblasts to provide a different collagen structure. Most of the research was performed on the human urinary bladder carcinoma cell line T24, which is one of the most widely used and extensively characterized bladder cancer cell lines. In addition, for the monitoring of DOX distribution in a microfluidic chip, human colorectal adenocarcinoma cells HT29 were used, because of their ability to remodel collagen. The cytotoxic effects of DOX were confirmed using a colony-forming assay, a live/dead cells assay, and ki67-staining. Metabolic effects of DOX were validated using a mitochondrial membrane potential assay and gene expression analysis.

## 2. Material and Methods

### 2.1. Cell Cultures

Human bladder carcinoma cell line T24, human colorectal cancer cell line HT29, and normal human skin fibroblasts (HuFb) were routinely grown in Dulbecco’s modified Eagle’s medium (DMEM; Gibco, Life Technologies, Carlsbad, CA, USA) supplemented with 10% fetal bovine serum FBS (HyClone, Logan, UT, USA), 2 mM glutamine (PanEco, Moscow, Russia), 10 mg/mL penicillin, and 10 mg/mL streptomycin in the incubator at 37 °C, 5% CO_2_, humidified atmosphere. Twice a week, cells were harvested using 0.025% trypsin-EDTA (Gibco, Life Technologies, Carlsbad, CA, USA).

### 2.2. Collagen-Based 3D-Model

A three-dimensional (3D) in vitro tumor model was obtained using previously developed protocol [21,22] with modifications of cell concentrations. Briefly, a solution of type I rat tail collagen (1.5 mg/mL) was mixed with a reagent mixture (10× Medium 199 (Gibco, Life Technologies, Carlsbad, CA, USA), NaOH, Na_2_CO_3_, glutamine and 1× HEPES) in the volume ratio 3.5:1. Then, a suspension of T24 or HT29 cells alone or with HuFb was added to the collagen gel in the ratio of 1:10. The total cell concentration was 2 × 10^5^ cells/mL with a final concentration of collagen gel of 1.2 mg/mL. In the case of the co-culture, T24 or HT29 cells and HuFb were mixed in the ratio of 1:1. The collagen gels were transferred in glass-bottomed FluoroDishes (WPI, China) for confocal microscopy or in the microfluidic chip and incubated overnight. T24 or HT29 cells seeded in the same amount but without collagen served as a control.

Since the morphology of T24 cancer cells and fibroblasts is similar, for their separation in the co-culture, HuFb were stained with the vital fluorescent dye CFSE (Thermo Fisher Scientific, Waltham, MA, USA) according to the manufacturer’s protocol.

### 2.3. Treatment with Doxorubicin

For the treatment, T24 cells were seeded either in 35 mm FluoroDishes in the amount 1 × 10^5^ cells/dish, or in 6-well plates–2 × 10^5^ cells/well, or in 96-well plates–2 × 10^4^ cells/well, depending on the experiment. Cells were treated with Doxorubicin (DOX, Teva, The Netherlands) in IC50 (50 µg/mL) or IC50/2 (25 µg/mL) concentration, determined by MTT assay. DOX was added to the cells 24 h after seeding. 

### 2.4. Design of the Microfluidic Chip (MFC)

The MFC, which was developed for this study, possessed two chambers: an inner chamber with hydrodynamic traps, and an outer chamber. The length × width × height of the inner chamber was 17 mm × 2.8 mm × 0.1 mm, while the length × width × height of the outer channel was 27 mm × 5.2 mm × 0.1 mm. The inner channel contained an array of 108 hydrodynamic traps with the following parameters: 0.07 mm (inner diameter) × 0.1 mm (outer diameter). Before the array of traps, four guiding lines were placed to homogenize flow distribution. The chambers were additionally separated by an array of pillars (Figure 1a). 

The mold for an MFC was made by etching the silicon substrate (depth of etching was 100 μm). In order to fabricate MFCs made of silicon elastomer PDMS, the “soft lithography” technique was used. For this, PDMS mixture of 1:10 ratio of curing agent to polymer base was prepared and poured onto a silicon mold. Then the uncured PDMS was degassed in a vacuum chamber for 20 min and cured at 80 °C during 2 h. Afterwards, the PDMS replica was sealed under a cover glass (170 μm) by oxygen plasma treatment. The geometry and the resulting MFC are presented in Figure 1.

### 2.5. Experimental Setup with MFCs for Drug Delivery Analysis

To detect doxorubicin spreading, we prepared MFCs with a 3D-collagen model, containing cancer cells T24 or HT29, co-culture of T24 or HT29 and HuFb without any cells, and MFCs with T24 or HT29 cells without collagen. For this, 20 µL of cell suspension in the growth media or cells mixed with collagen were immediately loaded in the central channel of MFCs and incubated for 30 min at 37 °C. Then, the fluid circulation system was turned on and fresh cultured medium was added at a flow rate of 90 µL/h. Observations were made at 24 h of cultivation. Doxorubicin was used in a concentration of 50 µg/mL, delivered to cells at a flow rate of 90 µL/h. Fluorescence was detected using a Leica DMIL fluorescence microscope (Leica, Wetzlar, Germany). Images were collected every 5 min during the first 30 min and then every 30 min during 2 h. For registration of fluorescence, a TX2 filter cube with excitation 560/40 nm and emission 645/75 nm was used.

### 2.6. Multiphoton Microscopy for NAD(P)H and DOX Imaging

Fluorescence intensity and lifetime images of the 3D cell cultures were acquired using an LSM 880 (Carl Zeiss, Jena, Germany) laser scanning microscope equipped with a FLIM module Simple Tau 152 TCSPC (Becker & Hickl GmbH, Berlin, Germany). A water immersion objective C-Apochromat 40×/1.2 NA W Korr was used for image acquisition. During image acquisition, the cells were maintained at 37 °C and 5% CO_2_. 

Two-photon fluorescence of NAD(P)H was excited with a femtosecond Ti:Sa laser (MaiTai, Spectra-Physics, Milpitas, CA, USA, repetition rate 80 MHz, pulse duration 140 fs) at 750 nm wavelength and registered in the range of 455–500 nm. The average power applied to the samples was ~6 mW, and the approximate rate of photon counting was 1–2 × 10^5^ photons/s. Image collection time was 60 s. Two-photon excitation of DOX fluorescence was performed at 780 nm and emission was registered using 690/50 filter. The photons were collected for 90 s. The average power applied to the samples was ~6 mW, and the approximate rate of photon counting was 1–2 × 10^5^ photons/s. In one-photon mode, DOX fluorescence was excited at 488 nm and registered in the range of 540–650 nm.

FLIM images of NAD(P)H and DOX were acquired sequentially from the same 5–7 randomly selected fields of view in each culture dish.

### 2.7. Single-Cell FLIM Analysis

To avoid inaccuracy associated with manual image processing and to obtain appropriate statistics, automated single-cell analysis of the FLIM images was performed. In the case of a large number of cells (>100), the manual segmentation procedure is labor-intensive, so here we used the segmentation algorithm described elsewhere [23]. Briefly, the analysis included two steps: (1) automatization and verification of cell segmentation with the cell cytoplasm and the nucleus being segmented in each cell; and (2) obtaining a fluorescence decay curve for each cell and fitting it with the appropriate decay model.

The cell segmentation algorithm was based on the U-net neural network pretrained on human colon adenocarcinoma cell images [23]. This model was retrained on 10 (3 for each model) manually segmented images (254 × 254 pixels each) of T24 cells, with augmentation (shift of the image in the ±50 pixels range, rotation in the 0–45° range, horizontal and vertical flip, zooming in 0–20% range) for 47 epoch while the loss metric stopped improving. Custom loss metrics were used. The Dice loss is widely used in medical image segmentation tasks to address data imbalance problems when searching for cell boundaries. Results of the boundaries and area detection for cells and nuclei were binarized and post-processed with the watershed algorithm as previously described [23]. The results represented the masks of single cells and nuclei. Implementation of the U-net model was performed in Keras library Python 3.1, and skimage and scipy libraries were used for image processing.

Since both DOX and NAD(P)H have rather weak fluorescence intensity in the cells, and the distribution of fluorescence lifetime parameters can vary inside a specific cell compartment, kinetics of all pixels in the ROI (individual cell cytoplasm or nucleus) were integrated. DOX fluorescence decay was then fitted with the monoexponential model to obtain decay curve parameters using scripts in the Python 3.1 using the IMFIT library. For the analysis of NAD(P)H fluorescence decay, the standard biexponetial decay model was used [24], and the short and long lifetime components (τ_1_ and τ_2_, respectively), the mean fluorescence lifetime (τ_m_ = (a_1_ · τ_1_ + a_2_ · τ_2_)/(a_1_ + a_2_)), and the relative amplitudes of the lifetime components (a_1_ and a_2_, where a_1_ + a_2_ = 100%) were estimated. The goodness of fit, the χ^2^ value, was 0.8–1.2. In a first approximation, the first (short, τ_1_) component of NAD(P)H fluorescence decay is attributed to its free state, associated with glycolysis, and the second (long, τ_2_) component to its protein-bound state associated with the mitochondrial respiratory chain. The detailed methodology can be found in Appendix A.

### 2.8. Second Harmonic Generation (SHG) Microscopy

The visualization of fibrillar collagen in 3D collagen-based models was performed using the SHG imaging option on the LSM880 microscope. The SHG signal from the collagen fibers was excited with a femtosecond Ti:Sa laser (MaiTai, Spectra-Physics, Milpitas, CA, USA) at a wavelength of 800 nm and detected in the range of 371–421 nm in the backward direction. The average power applied to the sample was ~12 mW.

### 2.9. Colony-Forming Assay

Firstly, T24 cells were seeded on 6-well plates in a concentration of 2 × 10^5^ cells per well in complete growth medium or in collagen gel. In 24 h DOX was added in IC50 concentration. After 24 h of incubation with DOX, T24 cells were trypsinized and counted, using an automated cell counter (T20 Bio-rad, Hercules, CA, USA). Some 4000 cells per well were plated in triplicate six-well plates in DMEM supplemented with FBS. Colonies were formed after 10 days. Colonies were fixed with ice-cold alcohol (95.0%), stained with crystal violet (0.5%) and observed using a Leica DMIL microscope (Leica, Wetzlar, Germany). For quantitative analysis, the crystal violet was eluted with alcohol and optical density was measured at 570 nm using a multimode microplate reader (Synergy Mx; BioTek Instruments, Winooski, VT, USA). 

### 2.10. MTT Assay

T24 cells were seeded in 96-well plates (5 × 10^3^ cells per well) and incubated for 24 h. Doxorubicin was added in concentrations of 10, 20, 50, 80, 100, 150, and 200 µg/mL. In 72 h of incubation, the MTT reagent 3(4,5-dimethyl-2-thiasolyl)-2,5-diphenyl-2H-tetrasolebromide (PanEco, Moscow, Russia) was added to the cells, according to the manufacturer’s protocol, and colorimetric analysis was performed at a wavelength of 570 nm using a multimode microplate reader (Synergy Mx; BioTek Instruments, Winooski, VT, USA). The percentage of viable cells was calculated as the ratio of formazan-stained treated cells to untreated control cells. For each concentration 10 internal replicates were performed. The IC50 was found to be 51 ± 4.5 µg/mL (Appendix A) and was used for all experiments.

To assess rapid changes, an MTT test was performed within 24 h after the addition of DOX. For this, AIM+ cells were plated on a 96-well plate in a concentration of 1 × 10^5^ cells without collagen and 2 × 10^5^ cells/well in collagen gel. In 24 h DOX was added in the IC50 concentration and an MTT test was performed at 30 min, 1, 5, 24 and 72 h. For each time point, 10 wells were used. Each experiment was carried out at least three times. 

### 2.11. Live/Dead Cell Assay

Cell viability was assessed with calcein and propidium iodide using live/dead cell double staining kit (Sigma, St. Louis, MO, USA) according to the manufacturer’s protocol. T24 cells were seeded in a 6-well plate with or without collagen, then DOX was added in IC50 concentration. Cells were stained after 24 and 48 h of DOX use, and the percentage of dead cells (PI-stained) from the total amount of cells was calculated. The fluorescent images were obtained using a wide-field Leica DMIL fluorescent microscope with a YFP ET filter (Ex: BP 500/20, Em: BP 535/30) for calcein and a TX2 green filter (Ex: BP 560/40, Em: BP 645/75) for PI.

### 2.12. Immunocytochemistry for ki-67

Immunocytochemistry for the cell proliferation marker ki67 was performed using primary rabbit antibody to ki-67 (MA5-14520, ThermoFisher) and goat anti-rabbit antibody, conjugated with Alexa 555 (ab150090, Abcam, Waltham, MA, USA). Cell membrane was permeabilized using 0.1 triton solution for 5 min, and nonspecific binding was blocked using block-solution with 1% BSA and glycine, 20 µM for 20 min. Primary antibody was diluted in block-solution and applied for 1 h at room temperature. Secondary antibody was diluted in PBS and applied for 30 min. Nuclei were counterstained using DAPI. Fluorescence was registered using a Leica DMIL microscope (Leica, Wetzlar, Germany) with TX2 filter cube with excitation 560/40 nm and emission 645/75 nm for Alexa 555 and CFP filter cube with excitation 436/20 nm and emission 480/40 nm for DAPI. For each time point a separate control was prepared. The fluorescence intensity was measured in the nuclei using ImageJ software. The mean fluorescence intensity was calculated for at least 30 cells.

### 2.13. Assessment of Mitochondrial Potential

The activity of mitochondria was evaluated using mitochondrial membrane potential indicator Image-iT™ TMRM Reagent (ThermoFisher, USA) according to the manufacturer’s protocol. Cells were seeded in 35 mm FluoroDishes with and without collagen, incubated overnight and treated with DOX (IC50) for 5, 24 or 48 h. Fluorescence of the indicator was registered using an LSM880 microscope (Carl Zeiss, Jena, Germany) upon excitation at 543 nm; detection range was 548–677 nm.

### 2.14. Isolation of RNA and Quantitative RT-PCR Analysis

T24 cells were seeded in 75 mm^2^ culture flasks (3 × 10^6^ cells/flask) or embedded in collagen gel: 3 mL of collagen + 4 × 10^6^ of T24 cells for monoculture and 2 × 10^6^ of T24 cells + 2 × 10^6^ of HuFb for co-culture. Before seeding, HuFb cells were stained with vital fluorescent dye CFSE (Thermo Fisher Scientific, USA), according to manufacturer’s protocol, in order to separate cancer cells from fibroblasts. T24 cells alone were incubated for 24 h and the T24 cells in collagen were incubated for 48 h. Afterwards, DOX was added in IC50 concentration into the test flasks, and growth media were changed in the control flasks. To prepare RNA samples, cells were trypsinized and collected after 24 and 48 h of DOX treatment for the assessment of expression of genes. Co-cultured cancer cells and fibroblasts were separated using a FACSAria III sorter (BD Biosciences, Franklin Lakes, NJ, USA). Cancer cells were collected into 15 mL falcon tubes filled with 5 mL of complete medium. Aliquots (5%–10%) of sorted cells were re-analyzed to check for sorting purity and were found to be in the range of 97.5%–99.8%. After sorting, the cells were washed with PBS and processed for RNA isolation.

The total RNA was isolated using RNA-Extran kit (Sintol, Russia). Before the reverse transcription reaction, the samples were treated with TURBO DNA-free™ Kit (Invitrogen, Carlsbad, CA, USA). Real-time PCR was performed using a CFX96 Real Time PCR system (Applied Biosystems, Waltham, MA, USA) and SYBR Green dye-based PCR amplification assay. PCR reaction contained 1-x GeneAmp PCR Buffer I (Applied Biosystems, Waltham, MA, USA), 250 µM of each dNTP, 0.5 nM of each primer (primer sequences are shown in Table 1), and 1 U of Taq M polymerase (Intifica, Saint Petersburg, Russia); total concentration of Mg^2+^ in the reaction was 3 mM and the reaction volume was 20 µL. The temperature profile of cycles: (1) 95 °C for 10 min (enzyme activation step); (2) 35 cycles of 95 °C for 15 s, 60 °C for 30 s, and 72 °C for 30 s; (3) hybridization 1 min 95 °C and 1 min 40 °C; and (4) melt curve analysis with measurements between 60 °C and 95 °C. The reaction efficiency was determined by LinRegPCR. Quantitative RT-PCR analysis was performed using CFX Maestro 2.3 software. The selection of reference genes was carried out using an integrated geNorm algorithm. The following reference genes were used: ABL1, SDHA1. The primer targets and sequences are listed in Table 1.

### 2.15. Statistical Analysis 

The mean values (M) and standard deviations (SD) were calculated to express the data of the drug cytotoxicity tests. Student’s t-test and one-way ANOVA with a Bonferroni post-hoc test, where appropriate, were used for data comparison, with *p* ≤ 0.05 considered statistically significant. 

For FLIM data, to calculate the statistical significance of the differences between the samples, the following procedure was used. First, outliers (data with parameter values below the <1 percentile and above the >99 percentile) were removed from the data set. Then, the distribution was tested for normality using the Shapiro-Wilk test (Shapiro’s implementation of the stats library in Python 3.1). Finally, the *p*-value for samples with different variances was calculated (ttest_ind of the stats library in Python 3.1).

## 3. Results

### 3.1. Effect of Collagen on DOX Delivery to Cancer Cells in MFC

The effects of collagen on drug distribution were investigated using type I collagen with a different structure. The structure of collagen was verified by SHG microscopy, an established optical technique based on the ability of centrosymmetric structures such as collagen fibers to reconvert two-photon excitation to form newly emitted photons with half of the wavelength [25]. It was shown that, in a 3D collagen-based model with T24 cancer cells, collagen was only poorly organized in thin, dispersed fibers, which became visible on SHG images in 48 h of cell culturing (Figure 1e). At an earlier time point (24 h), collagen in this system was unstructured and did not differ from control without cells. In the co-culture of T24 cells with fibroblasts, collagen was already organized in long, aligned fibers at 24 h, and then its structure did not change (Appendix A). 

Analysis of DOX distribution in MFCs loaded with collagen and cells was performed in dynamics using the intrinsic fluorescence of DOX. The drug reached T24 cells most quickly in the model without collagen, in 5 min. Unexpectedly, the unstructured collagen matrix in the model with T24 cells tended to be less permeable for DOX than in the co-culture with fibroblasts, where mature collagen fibers were presented due to the natural ability of fibroblasts to remodel the ECM. Fluorescence of DOX was detected in T24 cells in the co-culture in 20 min of the drug flow (Figure 1f). By 60 min of observation DOX reached the cells in all models without any changes for at least 2 h.

Similar experiments were performed on the HT29 cell line. Since HT29 cells themselves are able to remodel collagen gel with the fibers’ formation [21], DOX reached cancer cells in the MFC faster, in 20 min, compared to the model with T24 cells, where DOX reached cancer cells only by 60 min (Appendix A).

Thus, while the delivery of DOX to the cancer cells is easiest in the absence of any form of collagen, the unstructured form results in the most difficult penetration of the drug to the cells. In a situation when fibrillar collagen is present, the diffusion of the drug is facilitated compared to when only unstructured collagen exists around the cells.

### 3.2. Effect of Collagen on Cytotoxicity of Doxorubicin

To evaluate cytotoxic effects of DOX in the presence of collagen, several cell proliferation and viability assays were used. The colony-forming assay revealed that cells extracted from collagen after 24 h of DOX exposure formed the colonies more actively than cells exposed to DOX in the absence of collagen. The optical density of crystal violet was 2.2 times greater: 0.621 ± 0.099 for cells in collagen vs. 0.274 ± 0.049 (*p* = 0.000) for cells without collagen (Figure 2a,b). Since only a fraction of live cells retain the capacity to produce colonies, this result indicates lower toxic effects of DOX on cells cultured in collagen. 

The live/dead cell viability assay showed that the number of dead T24 cells significantly increased after 48 h of incubation with DOX, irrespective of the presence of collagen. However, in the case of cell-culturing in collagen, the percentage of dead cells was statistically lower, 28.3 ± 13.9% vs. 44.5 ± 18.3%, *p* = 0.006 (Figure 2c).

The immunocitochemical staining for ki-67, the marker of cell proliferation, at 5 h and 24 h of incubation with DOX revealed that expression of ki-67 in T24 cells without collagen statistically decreased, starting from 5 h of incubation (*p* = 0.020 and 0.000, correspondingly) (Figure 2c). T24 growing in collagen was slightly increased at 5 h (*p* = 0.01), and did not change at 24 h (*p* = 0.752).

Collectively, these results suggest that collagen, by limiting penetration of DOX to cancer cells in the 3D model, promotes cell survival upon DOX treatment.

### 3.3. Analysis of Doxorubicin Distribution by Multiphoton Microscopy

Analysis of fluorescence intensity of DOX in dynamics in T24 cells without collagen and in collagen-based 3D models revealed that DOX entered the cells more quickly if they were not embedded in collagen (Figure 3). DOX fluorescence was observed in both the cell cytoplasm and the nuclei already in 30 min incubation with the drug. In the cell cytoplasm, DOX was located mainly in the perinuclear zone and weakly co-localized with mitochondria, identified by the intense fluorescence of NAD(P)H (Manders overlap coefficient M1 0.297, Pearson correlation 0.186, Appendix A). Such distribution is typical for DOX, as demonstrated previously [17,18]. During the next 5 h, the signal intensity in the cytoplasm and in the nucleus increased, and then gradually decreased. After 48 h, fluorescence was preserved in both compartments, although in the nuclei it was weak. Cancer cells growing in unstructured collagen and in collagen structured by the fibroblasts showed generally slower intracellular accumulation of DOX. The signal of DOX was detected in the cytoplasm and in the nuclei in 1.5 h of incubation and gradually increased during 48 h. In the case of structured collagen, the intermediate values of DOX signals were recorded in the nuclei. Thus, the results of subcellular analysis of intrinsic DOX fluorescence were compatible with the data on DOX distribution in MFCs, where the highest rate of cellular DOX uptake was detected in the absence of a collagen matrix, and the slowest in the case of unstructured collagen. 

In an attempt to monitor DOX-target interactions, FLIM measurements were performed. In general, fluorescence lifetimes of DOX in the cells were longer than in DMEM growth medium (~1.70–1.9 ns vs. ~1.0 ns), and within a cell they were slightly longer (by 3–10%) in the cytoplasm than in the nucleus, which is consistent with previous research [24,26]. The lowest values of fluorescence lifetime (~1.7 ns) were recorded in the nuclei of T24 cells from 1.5 h to 24 h in all the systems. In the nuclei of cells without collagen, the fluorescence lifetimes in this period were insignificantly lower compared to the cells in collagen. According to other studies, decrease in DOX fluorescence lifetime is an indicator of its binding to chromatin [24,26]. Therefore, the dynamics of the binding did not depend on the drug uptake and was generally similar for the models with and without collagen.

In the cytoplasm, the fluorescence lifetime of DOX in T24 cells in all cases also decreased from 1.9 ns to 1.8 ns during 5 h of incubation, and then increased back to the baseline. 

Collectively, the results of multiphoton microscopy of DOX in fluorescence intensity and lifetime modes point to the significant variations in the level of cellular uptake depending on the presence of collagen, whereas drug–target engagement did not depend on it.

### 3.4. NAD(P)H FLIM of T24 Cells under Doxorubicin Exposure

The use of the fluorescence lifetime of NAD(P)H as a metabolic metric is based on its dependence on the binding to proteins. NAD(P)H in its free state has a stably short lifetime, 0.4–0.5 ns, and is typically assigned to the glycolysis process. Bound state of NAD(P)H with longer lifetime, typically ~2–3 ns, is associated mainly with mitochondrial complexes [16,27]. Therefore, the relative changes in the metabolic fluxes can be assessed either from average-weighted (mean) lifetime or from the contribution of the short (a_1_) and long (a_2_) lifetime components to the fluorescence decay.

Fluorescence lifetime values of NAD(P)H, τ_1_ and τ_2_, under baseline conditions were calculated to be ~0.38 ns and ~2.7 ns (Appendix A). Upon treatment with DOX, both lifetime values remained unchanged (deviations did not exceed 5% of their baseline values), which means that no significant alterations in the cellular environment (viscosity, pH, etc.) and composition of NAD(P)H-binding enzymes accompany the cell response.

It should be noted that NAD(P)H fluorescence lifetime parameters were different for cancer cells growing in different conditions (Figure 4 and Appendix A). The raw SPCImages with higher magnification are shown in Appendix A. In the presence of collagen and fibroblasts, the mean lifetime τ_m_ was statistically lower (0.75 ns vs. 0.89 ns) and the relative contribution of free NAD(P)H a_1_ in T24 cells was higher compared to the model without collagen (83% vs. 77%). In cells embedded in unstructured collagen, a tendency to lower τ_m_ and higher a_1_ with a significant variability between cells was also noted. These changes in τ_m_ and a_1_ parameters typically indicate a shift to a more reduced cellular redox state, e.g., due to inhibition of OXPHOS or enhanced glycolysis [28]. The observed features of the cells in collagen gel correlated with their decreased metabolic activity and proliferation rate. 

After adding DOX, a rapid modification of NAD(P)H fluorescence lifetime parameters was observed. τ_m_ gradually decreased during 1.5 h from 0.89 ns to 0.75 ns and remained decreased until 24 h in the cytoplasm of T24 cells without collagen (Figure 4. Correspondingly, a_1_ value increased from 77% to 82%. In the cells grown in unstructured collagen, τ_m_ and a_1_ insignificantly fluctuated during 5 h; by 24 and 48 h a_1_ was lower than in control (0 h), which points to their more oxidative status. In the model with fibroblasts, τ_m_ and a_1_ showed opposite changes compared to the cells without collagen: τ_m_ increased from 0.75 ns to 0.89 h and a_1_ decreased from 83% to 77% after 0.5 h and remained at these levels up to 48 h. In the nuclei, the changes in NAD(P)H lifetime parameters were similar to those in the cytoplasm. 

Therefore, the dynamics and type of metabolic perturbations in cancer cells depended on the presence of collagen and fibroblasts. The cells in standard culturing conditions without a collagen matrix were initially in a more oxidized state and exhibited rapid changes toward a more glycolytic state at the early time points (<24 h), while the cells in collagen initially had more glycolytic metabolism and showed a shift to oxidative status upon treatment with DOX.

To test whether the different metabolic shift in the cells in collagen was due to lower drug concentration, we additionally performed the experiment on pure T24 cells using twice-lower dose of DOX (IC50/2). It was found that the lower dose induced changes in cellular metabolism similar to the conventional IC50 dose (Appendix A). At that level, fluorescence intensity of DOX was comparable with that in collagen-embedded cells (Appendix A). These results suggest that different metabolic effects of DOX on cells with and without collagen are a consequence of their initially different metabolic status rather than of different drug doses delivered to the cells.

### 3.5. Metabolic Activity of T24 Cells after Treatment with Doxorubicin

In order to determine the origin of metabolic rearrangements detected by NAD(P)H FLIM, mitochondrial activity was assessed using TMRM, the indicator of mitochondrial membrane potential (MtMP). It should be noted that the initial fluorescence intensity of TMRM was significantly lower in T24 cells grown in collagen compared to T24 cells alone, 15.6 ± 3.3 vs. 25.7 ± 6.4 a.u. (*p* = 0.000), which indicates their lower MtMP. Treatment with DOX resulted in a slight but statistically significant increase in TMRM fluorescence after 5 h in monoculture of T24 cells in collagen and after 5 and 48 h in co-culture with HuFb in collagen. In T24 cells cultured without collagen, the TMRM indicator revealed a small decrease in MtMP, starting from 5 h of DOX exposure, with a dramatic decrease at 48 h (Figure 5a).

The early changes in mitochondrial activity were additionally confirmed by the MTT assay, which demonstrated a higher activity in T24 cells growing in the collagen matrix in 5 h after addition of DOX compared with untreated control and cells in collagen. The photographs of formazan distribution showed the mitochondrial fission in the treated T24 cells without collagen and the mitochondrial fusion in cells seeded in collagen, reflecting, correspondingly, decreased and increased OXPHOS (Appendix A).

To clarify the molecular mechanisms of metabolic changes in T24 cells, the expression level of the key metabolic genes was analyzed using RT-PCR (Figure 5b). In T24 cells growing in collagen matrix, the levels of HK1 and HK2, the first step in most glucose metabolism pathways, were lower, and the levels of LDHA, associated with glycolysis, and G6PD, associated with the pentose phosphate pathway, were higher, compared to T24 cells without collagen. Treatment with DOX led to a significant increase in the expression of LDHA and PDP1 in T24 cells without collagen, while in T24 cells in collagen, LDHA level increased to a lower degree and the expression of PDP1 decreased. The expression of G6PD increased in the cells without collagen and decreased in the cells in collagen. The relatively stable expression of OGDH, the TCA cycle enzyme, reflected the unaltered activity of the TCA cycle in all models.

Overall, these results are consistent with NAD(P)H FLIM that established different metabolic responses of cancer cells to DOX depending on the presence of collagen and fibroblasts in their microenvironment.

## 4. Discussion

The influence of the extracellular matrix (ECM) on the response of a tumor to chemotherapy has been a subject of active research for many years. DOX remains one of the most effective and widely used chemotherapy agents, but resistance of cancer cells to its action severely restricts its application [10]. Recent data suggest that proteins of the extracellular matrix, collagen deposition, and its specific organization in the tumor, promote resistance to DOX [29,30]. However, the mechanisms of DOX resistance associated with ECM are still poorly understood. In our study, we used a model system, based on 3D-collagen matrix and MFC, for analysis of DOX delivery and distribution in cancer cells, and applied multiphoton FLIM technique to follow simultaneously the drug-target interaction (assessed from DOX fluorescence lifetime) and metabolic status of cancer cells (assessed from NAD(P)H fluorescence lifetime).

To investigate the effects of different ECM properties on drug response, appropriate models are needed. The most common in vitro models for studying drug penetration are multicellular spheroids, multilayered cell cultures, and microfluidics devices [31,32,33]. However, in most systems ECM compounds are normally not included. The influence of ECM on cancer cells can be investigated by in vivo tests using fluorescent dyes, such as Hoechst, or drugs with intrinsic fluorescence, such as DOX or Etoposide [24,34]. However, in this case experimental conditions are poorly controlled and any assessment of contribution of individual factors in the ECM is limited. Our proposed model mimics blood flow by using MCFs and the ECM structure in a realistic environment on the base of the major ECM protein collagen and fibroblasts as a structuring agent.

We found that long-term cell viability after DOX exposure was higher for cells growing in collagen than for cells without a collagen matrix, which can be explained by various factors. First, lower amounts of DOX entering cells through the collagen matrix could lead to restricted cytotoxicity. Second, decreased proliferating activity of the cells in collagen could be another factor, as chemotherapeutic drugs, including DOX, mainly affect proliferating cells [35]. Third, metabolic rearrangements in cancer cells in the presence of fibroblasts and collagen, specifically a shift to anaerobic glycolysis, can contribute to cell viability. Finally, cell-to-ECM interactions through the adhesion molecules integrins can contribute to the resistance to DOX by regulating signaling networks, such as FAK/Src, NF-kB, MARK, and PI3K, that support cancer cell proliferation, survival, and invasion [29]. 

The advantage of DOX in studying the effects of ECM on drug efficacy is in its intrinsic fluorescence, which opens the possibility to monitor its accumulation and subcellular distribution using fluorescence microscopy. Doxorubicin has an absorption peak in the range of ∼480 to 500 nm (λ max. ex. 490 nm) and a broad emission in the range of ∼520 to 720 nm (λmax. em. 590 nm) [36]. Analysis of DOX distribution in MFCs by its fluorescence revealed that a totally unstructured collagen matrix was less permeable for the drug than collagen comprising mature fibers formed by fibroblasts. Collagen fibers make tracks in the ECM that on the one hand support tumor metastasis, but on the other, facilitate drug delivery. Enhanced tumor tissue permeability for anticancer drugs after ECM modification, using enzymes such as collagenases and hyaluronidases, has been demonstrated in numerous studies [7,8,9]. This is, in principle, in line with our data on most effective penetration of DOX in the model without collagen. 

At the cellular level, DOX fluorescence was observed both in the nuclei and in the cytoplasm of T24 cancer cells, while not co-localizing with mitochondria. It was demonstrated earlier on breast cancer cell lines BT-474 and MCF7 that DOX enters the nuclei of cancer cells without entering mitochondria [17,18]. At the same time, numerous studies have reported that DOX accumulates in the mitochondria and perturbs mitochondrial bioenergetics and function, thus considering mitochondria as a target of DOX toxicity [11,12]. Our observations of mitochondrial redox state using NAD(P)H fluorescence revealed that changes in the protein-bound NAD(P)H fraction occurred early (within 30 min) in the course of treatment with DOX, which point to the direct interaction of DOX with mitochondria rather than indirect via nucleus. The absence of DOX fluorescence in mitochondria can be explained by rapid bioactivation of DOX by mitochondrial complex I or by cytochrome P450 with the production of a semiquinone radical and, consequently, the loss of fluorescence. 

FLIM has proven to be a valuable tool to dynamically monitor the DOX–DNA intercalation as it is accompanied by a decrease in the DOX fluorescence lifetime [24,26]. Our fluorescence lifetime measurements of DOX corroborate previous findings that show lifetime elongation of DOX in cells compared to solution [24]. In a study by Chen et al., DOX fluorescence lifetime in the nuclei of HeLa cells decreased rapidly during the first 2 h of treatment [26]. Similarly, in our experiments DOX lifetime decreased by 1.5 h and remained low until 24 h irrespective of the presence of collagen, indicating a very quick and effective binding. A FLIM study of DOX binding to chromatin in vivo was performed by Sparks et al. in a mouse model of ovarian cancer metastases. The authors observed considerable variations in DOX–chromatin binding between different tumor nodules and within the same nodules, as well as depending on the drug delivery route [37]. However, the reasons for this heterogeneity have been unclear so far. The present study suggests that different rates of drug uptake, distinct collagen structure, and the local variations in the metabolic state of the cells are unlikely to be responsible for the differences in drug-target engagement.

Recent data relating to mechanisms of DOX action suggest that they are not limited to DNA damage alone, but also involve mitochondrial damage [11,12]. DOX-induced mitochondrial perturbations are usually considered a cause of its remarkable cardiotoxicity, a serious side effect of this drug [38,39]. Several studies have shown that DOX inhibits mitochondrial respiration via inactivation of complexes I, II, and IV of the electron transport chain, and also phosphate carrier, due to binding to cardiolipin in the inner mitochondrial membrane [40]. Meanwhile, the existing literature about the effects of DOX on mitochondrial function is quite controversial and suggests that the effects depend on many factors, including drug dose, cell type and their respiratory capacity, availability of oxygen, substrates, and other factors of cellular metabolism [41,42]. Since DOX fluorescence is spectrally distinct from the autofluorescence of NAD(P)H, there is an opportunity to probe intracellular distribution and binding of DOX simultaneously with the metabolic state of the cells. To the best of our knowledge, the combined imaging of DOX and NAD(P)H has not been realized so far.

In this study we detected different changes in the mitochondrial redox state of T24 cells in the absence and presence of collagen. Metabolic FLIM revealed a rapid decrease of the NAD(P)H mean fluorescence lifetime due to an increase in the fraction of free NAD(P)H in T24 cells growing without collagen, likely associated with DOX-induced mitochondria damage and testifying to a shift toward a more reduced (glycolytic) metabolism. However, upon prolonged incubation with DOX, their optical metabolic metrics returned to the control, which could be explained by the fact that the remaining viable cells became, in general, metabolically inactive. NAD(P)H FLIM has been applied to monitor metabolic response to chemotherapy in numerous studies, including ours [23,43,44,45]. Interestingly, drugs with different action mechanisms (e.g., cisplatin, 5-fluorouracil, taxol, irinotecan) showed similar effects in terms of NAD(P)H fluorescence decay parameters—a shift to a longer mean lifetime and increased contribution of protein-bound fraction, which correlated with a decreased cell proliferation rate. The effect of DOX was principally different and resembled that of the mitochondrial inhibitors [46,47].

It should be noted that T24 cells in the absence of collagen initially had more oxidative metabolism compared to those in the collagen matrix, according to NAD(P)H FLIM data and the mitochondrial membrane potential assay. It was widely shown that inhibition of glycolysis resulted in enhanced sensitivity to DOX [48]. Vice versa, many cellular processes are more resistant to DOX under glycolytic conditions, demonstrated in the yeast phenomic experiments that mimicked the Warburg effect [41]. An early increase (starting from 15 min after the addition of DOX) in ROS production, which mainly takes place in mitochondria, was demonstrated in several studies and correlated with DOX toxicity [20,49]. Moreoever, it was shown that cells with an intact electron transport chain growing under normoxic conditions were more sensitive to DOX than cells with compromised electron transport capabilities [41,42,50]. Inhibition of the complex I by DOX causes a reduction in the release of mitochondrial cytochrome c, which contributes to the inhibition of caspase activation and apoptosis [49]. In turn, the addition of mitochondrial inhibitor dinitrophenol partially mitigated proliferation arrest induced by DOX in MCF7 breast cancer cells [42]. All these findings may be explained by decreased activation of DOX in impaired mitochondria as the quinone in the DOX structure is oxidized by the mitochondrial complex I (NAD(P)H-oxidoreductases) and NADPH oxidases (NOXs), resulting in ROS production [51]. 

In collagen-based models the cells were initially in a more reduced state and became more oxidative upon DOX treatment. Gene expression analysis showed activation of glycolysis (LDHA) and general decrease in metabolic activity (HK1, HK2) in T24 cells in collagen, which correlated with higher contribution of free NAD(P)H assessed from FLIM. This is in agreement with literature reporting on a glycolytic phenotype of cancer cells in collagen-rich ECM due to hypoxic conditions [52,53]. It is notable that in the case of co-culture with fibroblasts, the initial differences with monoculture were more pronounced. Previously, we have demonstrated on HeLa cells that interaction with fibroblasts could have a profound effect on their metabolism with the induction of a shift to glycolysis [22]. At long-term exposure, mitochondrial activity seems to recover—the decrease in the relative contribution of free NAD(P)H was detected, which indirectly indicates a shift to a more oxidative state. This result is consistent with the study by Achkar et al., where inhibition of glycolysis in response to DOX was found in breast cancer in the OVO-model [14]. Alam et al. and Wallrabe et al. demonstrated increased enzyme-bound NAD(P)H fraction in prostate cancer cells after treatment with DOX [19,20], similar to our observation of T24 cells growing in collagen. Prostate cancer cells have a significantly decreased mitochondrial function and reduced OXPHOS due to truncated Krebs cycle and increased accumulation of citrate, owing to prostate physiology [54,55]. In our study, lower mitochondrial activity of cancer cells in collagen and in the presence of fibroblasts correlated with higher cell viability. It was previously shown that metabolic effects of DOX are dose-dependent. For example, treatment of MCF7 breast cancer cells with a low dose of DOX, which was not acutely toxic, resulted in significantly increased mtDNA copy number indicating metabolic accommodation to the damage caused by DOX [42]. 

The limitation of NAD(P)H-based FLIM as an approach to characterize metabolic processes in cells is that it provides information about relative changes in the concentrations of free and bound NAD(P)H pools, but does not report on the specific metabolic perturbations that caused the changes. The changes in the values of free and bound NAD(P)H fractions can be attributed to the alterations in glycolysis, OXPHOS, glutaminolysis, fatty acid oxidation, and synthesis [56]. Of these, changes in the balance between the relative levels of glycolysis and OXPHOS are thought to be the most prevalent metabolic adaptation of cancer cells in response to hypoxia, oxidative stress, and proliferative needs [57]. Meanwhile, the possibility to non-invasively monitor the metabolic state of cells on a label-free basis with a sub-cellular resolution, and combine the detection of cellular autofluorescence with other fluorophores with different spectral characteristics, makes FLIM microscopy a powerful technique in drug studies. A detailed biochemical and molecular characterization of the cellular response to DOX in the collagen environment will be performed in our future research. 

We have demonstrated in vitro, in well-controlled and realistic conditions of 3D models, the role of ECM structure in DOX delivery and efficacy against cancer cells. For the first time, using a combination of SHG, DOX FLIM, and NAD(P)H FLIM options in multiphoton microscopy, we have related the levels of drug accumulation, drug-target engagement, and metabolic perturbations in cancer cells to their collagen environment. We should note that the obtained results are valid for T24 and HT29 cell lines, which the former does not and the latter moderately modifies collagen structure in the absence of fibroblasts. Obviously, the situation is more complex in a solid tumor as its development is accompanied by various ECM-remodeling events driven by bidirectional communication between cancer and stromal cells. This results in a highly changeable and nonuniform ECM. In terms of metabolic rearrangements induced by the drug, the initial bioenergetic profile of cancer cells plays a role. Apart from collagen and cancer-associated fibroblasts, a variety of other factors affect cancer cell metabolism in a tumor. Therefore, a high degree of metabolic heterogeneity is expected in response to the treatment. These and other questions remain open and hopefully will be addressed in the future.

## 5. Conclusions

Despite significant progress in the understanding of the role of ECM in tumor progression, the effects of collagen on the response of tumor cells to chemotherapy have been elucidated poorly. Our study, for the first time, highlighted the links between the structure of collagen matrix, cellular metabolism, and cytotoxicity of DOX in bladder cancer cells. On the one hand, collagen served as a protective barrier for drug distribution to tumor cells. In general, collagen significantly slowed down drug penetration and decreased the amount of the drug delivered into cells, although the fibrillar structure formed by the fibroblasts improved drug delivery to a certain degree compared with unstructured collagen. On the other hand, cellular metabolism shifted to a more glycolytic form in the presence of collagen and fibroblasts, which could be an additional factor of resistance to DOX. In more oxidative cells growing in the absence of collagen, treatment with DOX resulted in a significant and irreversible impairment of mitochondrial respiration, which likely contributed to the greater toxicity of DOX. Our results suggest that the complexity and variability of collagen organization in tumors could be one of the reasons for heterogeneous and sub-optimal responses of tumor cells to DOX.

## Figures and Tables

**Figure 1 cancers-14-05487-f001:**
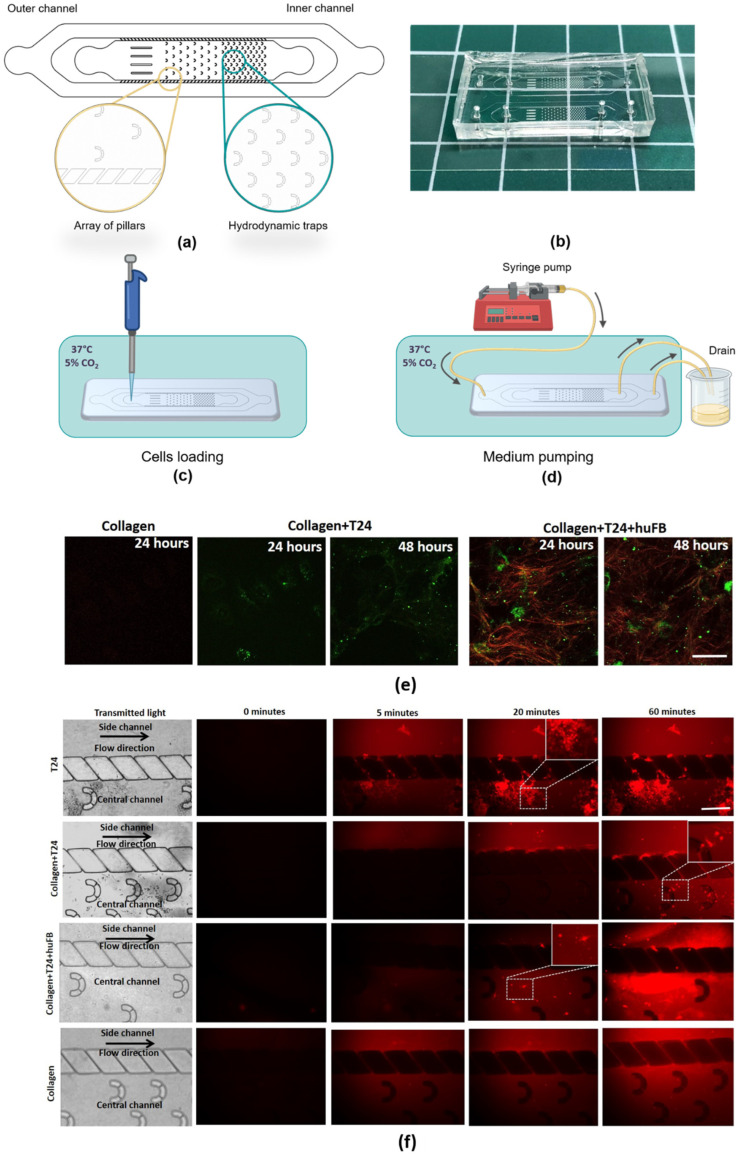
Modeling of DOX delivery to cancer cells in the presence of collagen using MFCs. (**a**) Geometry of the MFC. (**b**) Resulting MFC. (**c**) Scheme of the experiment with microfluidic chips. Cells loading into the MFC channel and further storage under CO_2_ and 37 °C for cell culture adhesion. (**d**) Medium pumping by a syringe pump connected to the MFC with a tubing system that leads the flow to the drain. (**e**) Representative SHG images of collagen seeded with T24 cancer cells or co-culture of T24 and human fibroblasts HuFb or without any cells. Red—SHG signal from collagen, green—autofluorescence of cells. Scale bar: 50 µm, applicable to all images. (**f**) Images of MFCs in transmitted light and representative time-lapse images of DOX fluorescence in different models. DOX was added in the concentration of 50 µg/mL (IC50). The areas of DOX uptake into T24 cells are shown in the dashed squares and enlarged in the right upper corner of the images. Red—fluorescence of DOX. Scale bar: 400 µm, applicable to all images.

**Figure 2 cancers-14-05487-f002:**
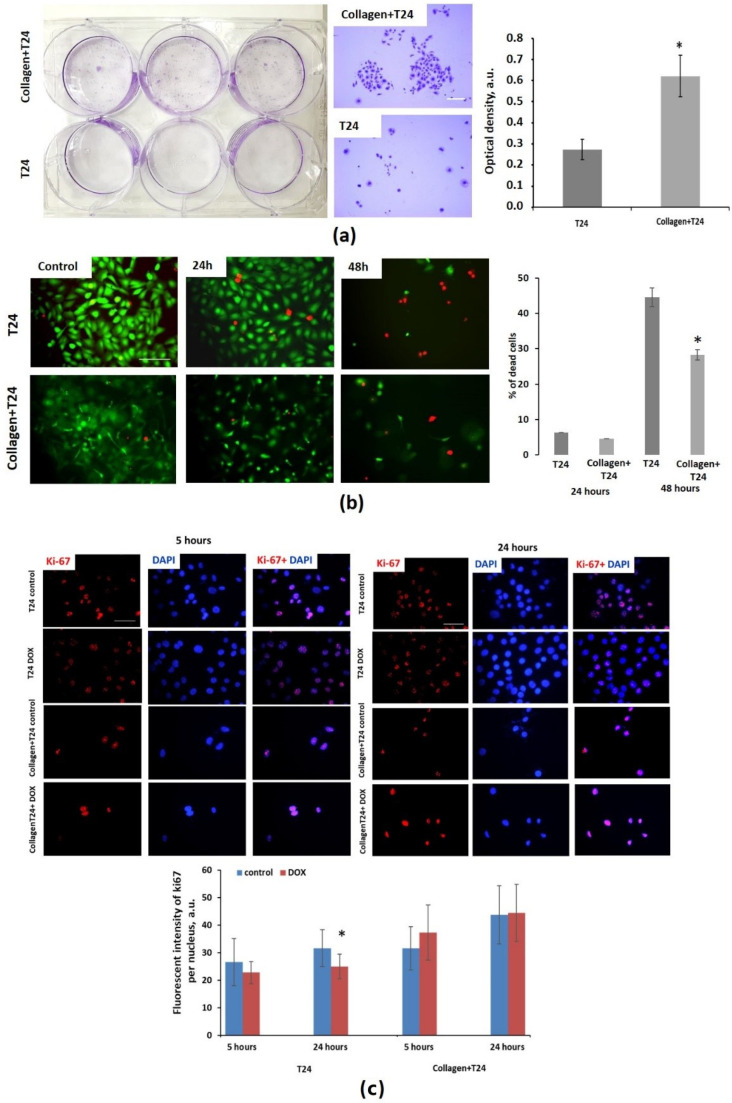
Cytotoxic effects of DOX on T24 cells in the presence and absence of collagen. (**a**) Colony- forming assay, picture of plate and corresponding microphotographs. Scale bar: 200 µm. Optical density of the eluted crystal violet dye. (**b**) Representative fluorescence images of cells stained with live/dead cell assay kit (green—live cells, red—dead cells). Dead cells count after 24 and 48 h of DOX exposure with and without collagen. Concentration of DOX was 50 µg/mL (IC50). (**c**) Expression of ki-67 in T24 cells seeded with and without collagen after 5 and 24 h of DOX exposure. Normalized fluorescence of ki-67. Mean ± SD, n = 30. * Statistically significant difference from T24 cells without collagen (**a**,**b**) or from untreated control (**c**) in the same group.

**Figure 3 cancers-14-05487-f003:**
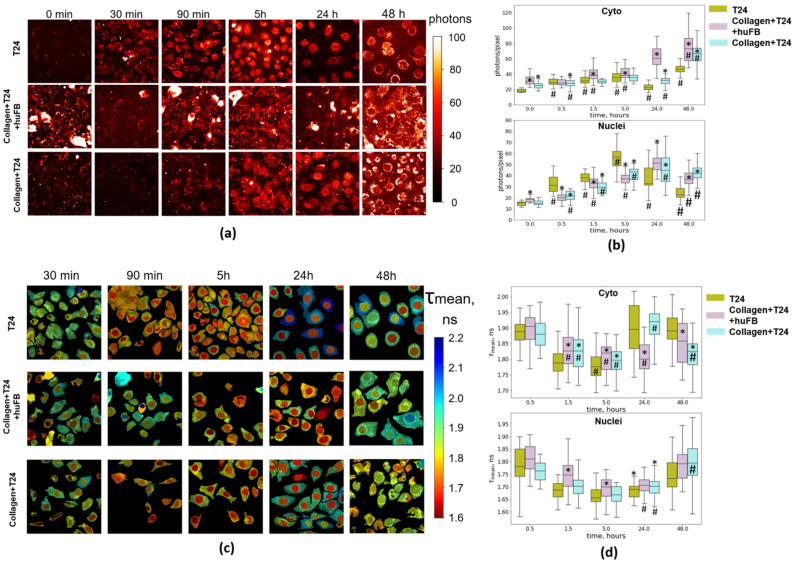
Fluorescence intensity and lifetime imaging of DOX in T24 cells in the absence and presence of collagen using multiphoton microscopy. (**a**) Representative fluorescence images of cells upon incubation with DOX. White arrow shows the fibroblast stained with fluorescent dye CFSE. (**b**) Analysis of fluorescence intensity of DOX in the cell cytoplasm and in the nuclei. Boxes are quartiles, whiskers are minimums and maximums. (**c**) FLIM images of cells upon incubation with DOX. The colors represent the mean fluorescence lifetimes evaluated from the integrated signal over the single-cell cytoplasm or the nuclei. Excitation: 780 nm, registration: 540–650 nm. Incubation time is indicated on the images. (**d**) Analysis of fluorescence lifetimes of DOX in the cell cytoplasm and in the nuclei. Boxes are quantiles, whiskers are minimums and maximums. Scale bar is 50 µm, applicable to all images. Concentration of DOX was 50 µg/mL (IC50). * Statistically significant difference from T24 cells without collagen at the selected point. # Statistically significant difference from the control value.

**Figure 4 cancers-14-05487-f004:**
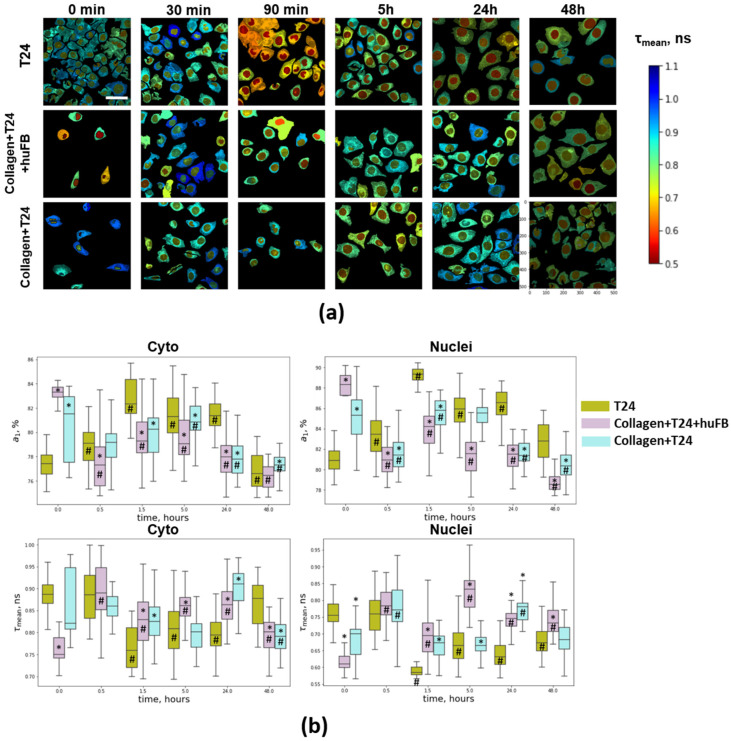
Fluorescence lifetime imaging of NAD(P)H in T24 cells in the absence and presence of collagen using multiphoton microscopy. (**a**) Representative FLIM images of cells upon incubation with DOX. The colors represent the mean fluorescence lifetimes evaluated from the integrated signal over the single-cell cytoplasm or the nuclei. Incubation time is indicated on the images. Excitation: 750 nm; registration: 455–500 nm. Scale bar is 50 µm, applicable to all images. Concentration of DOX was 50 µg/mL (IC50). (**b**) Analysis of relative contribution of free NAD(P)H a_1_ and fluorescence lifetimes τ_m_ of NAD(P)H in the cell cytoplasm and in the nuclei. Boxes are quartiles, whiskers are minimums and maximums. * Statistically significant difference from T24 cells without collagen at the selected point. # Statistically significant difference from the control value.

**Figure 5 cancers-14-05487-f005:**
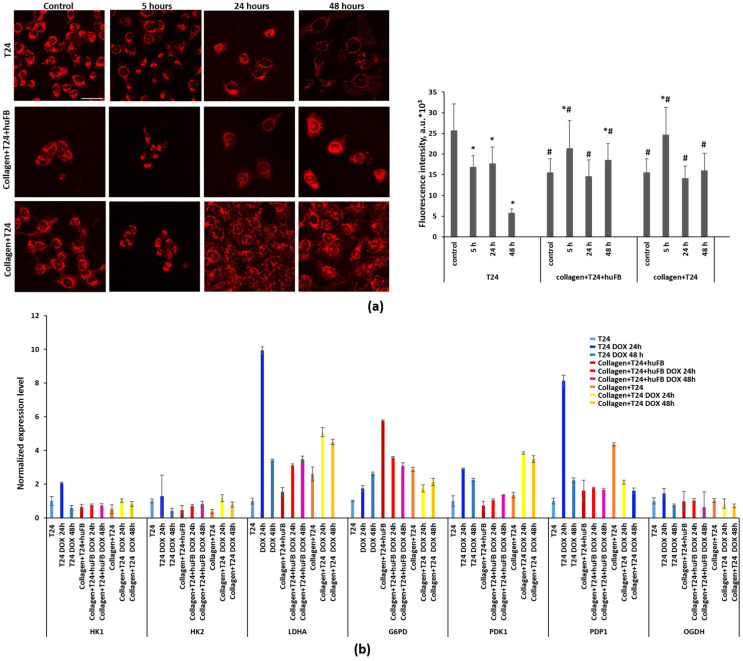
Metabolic activity of T24 cells after treatment with DOX in the absence and presence of collagen. (**a**) Visualization of mitochondrial membrane potential using fluorescence microscopy and TMRM indicator. Scale bar is 50 µm, applicable to all images. Fluorescence intensity of TMRM in cells. Mean ± SD. * Statistically significant difference from the control value. # Statistically significant difference from T24 cells without collagen at the selected point. (**b**) Analysis of the expression level of metabolic genes HK1, HK2, LDHA, G6PD, PDK1, PDP1, OGDH using RT-PCR. Expression was normalized to the values for the T24 cells. Concentration of DOX for the cell treatment was 50 µg/mL (IC50).

**Table 1 cancers-14-05487-t001:** Gene list and their primer sequences for RT-PCR.

Primer Target	Description	Primer Sequence (5′→3′)
HK1	Hexokinase-1	Forward Primer-CACCTGTGAGGTTGGACTCA Reverse Primer-CCACCATCTCCACGTTCTTC
HK2	Hexokinase-2	Forward Primer-GAGTTTGACCTGGATGTGGTTGC Reverse Primer-CCTCCATGTAGCAGGCATTGCT
PDK1	Pyruvate dehydrogenase kinase	Forward Primer-CCGCTCTCCATGAAGCAGTT Reverse Primer-TTGCCGCAGAAACATAAATGAG
LDHA	Lactate dehydrogenase A	Forward Primer-AGCCCGATTCCGTTACCT Reverse Primer-CACCAGCAACATTCATTCCA
G6PD	Glucose-6-phosphate dehydrogenase	Forward Primer-CTGTTCCGTGAGGACCAGATCT Reverse Primer-TGAAGGTGAGGATAACGCAGGC
OGDH	2-Oxoglutarate dehydrogenase	Forward Primer-GAGGCTGTCATGTACGTGTGCA Reverse Primer-TACATGAGCGGCTGCGTGAACA
ABL1	Tyrosine-protein kinase	Forward Primer-CCAGGTGTATGAGCTGCTAGAG Reverse Primer-GTCAGAGGGATTCCACTGCCAA
SDHA1	Succinate dehydrogenase Complex flavoprotein Subunite A	Forward Primer-GAGATGTGGTGTCTCGGTCCAT Reverse Primer-GCTGTCTCTGAAATGCCAGGCA
PDP1	Pyruvate dehydrogenase Phosphatase catalytic Subunit 1	Forward Primer-TTCTGGAGCCACTGCTTGTGTG Reverse Primer-ACAGCGTGACTGCTGACCATGA

## Data Availability

The data that support the findings of this study are available from the corresponding author upon reasonable request.

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
