# Peer review of "Effect of Collagen Matrix on Doxorubicin Distribution and Cancer Cells’ Response to Treatment in 3D Tumor Model"

_cancers, 2022, doi:10.3390/cancers14225487_

Round 1

Reviewer 1 Report

The researchers presented a series of experiments in triplicate of bladder carcinoma (T24) and colorectal cancer (HT29) cell lines, which suggest that collagen may limit the penetration of DOX to cancer cells in a 3D model. The researchers suggest that this mechanism amongst others promotes cell survival under DOX treatment.

The experiments focus on cytotoxicity of DOX and do not consider any other experimental mechanisms, such as treatment-generated resistance. The researchers do mention that other mechanisms can contribute to resistance and provide suitable references.

The researchers conclude that collagen organisation in the tumours can contribute towards heterogeneity and sub-optimal DOX responses and that variations in collagen structures and the metabolic state of cells are unlikely to be responsible for differences in drug-target engagement in T24 cells.

The paper is very well written an engaging. The experiments are sound, and the results clearly shown. The supplemental figures were essential to understand the results. We appreciate the researchers stating that the conclusion are supported by the experiments, but there are still open questions that hopefully will be addressed in the future. We would suggest the authors emphasise that these results are valid for T24 and HT29, and perhaps offer a short rationale for the selection of these.

Typos:

line 205- Extra space after “Python 3.1”

line 429- missing “tm” symbol

line 485- should “model systems” be singular?

Figure 3 is shifted in the page

Reviewer 2 Report

As a whole the manuscript is interesting. Using the combination of SHG, DOX FLIM and NAD(P)H FLIM in multiphoton microscopy provides positive results. However, additional classical methodologies (i.e. western blot) are required to validate some results (i.e. metabolic perturbations in cancer cells). More experimental evidence is needed (i.e. explore changes in mitochondrial membrane potential, release of cytochrome c and others..).

Additional comments:

In page 2, references are needed for the following sentence": In addition, the ECM strongly affects transport mechanisms, metabolism, oxygenation and immunogenicity of solid tumors, and thus, along with biological behavior of tumor cells, it also regulates their response to therapy."

In page 8, the presence of mature collagen fibers in the samples with fibroblasts must be better explained.

Author Response

Plrease see the attachment

Reviewer 3 Report

This manuscript is overall well written. It addresses a clear and important issue, asks pertinent questions and uses relatively appropriate methods. However, it has some issues, and some of the data needs further development. More specifically:

Figure 1 and 3 can be combined:

            Material and methods 2.4: What do array of pillars and hydrodynamic traps do to cells? Please explain method also from cells point of view.

            Material and methods 2.5 line 155: 20ul of what mixture added? I am guessing this is cell mixture. Were cells allowed to settle in MFC after 30minutes of incubation before introduction of Doxorubicin? Can cells attach to any surface in MFC or are they in suspension?

 Figure 2: can be pushed to supplemental data as it was previously published.

 Figure 3 a: What is rationale of using T24 cells? It would be validating to see 1 or 2 more bladder cancer cell lines in parallel with T24.  

 What is rationale of selecting 24hr time point for most studies? Previous studies have shown that increase in collagen density was seen at day 5.

Also, if flow direction is from array of pillars to hydrodynamic traps, T24 image seems to be at arrays of pillars and CollagenT24 image seems to be at hydrodynamic traps. Aren't they supposed to be taken at relatively same space? Can this effect the time point of dox delivery to cells?

 Figure 4 a&b: For colony formation assay why cells were allowed to grow only for 10days? If they had allowed colonies to form little longer, they would have had more bigger and stronger colonies in collagen+T24 group and optical intensity value would have been significant.  Please add doxorubicin and concentration used on figures.

 Figure 4c: Again, 24hr time point is very short for doxorubicin, instead 48hr or 72hr time point should be used. Dead cells would be lost during trypsinization, therefore instead of trypan blue quantification BrdU assay or acridin orange/PI stain would give significant results.

 Figure 4d: Why is density of cells low in collagenT24 groups? Authors should provide pictures of relatively same number of cells between both groups. Again, 48hr or 72hr time point would have given significant results. Authors should provide higher magnification pictures.

 Figure 6: MTT assay here is unnecessary, I suggest authors perform mechanistic studies.
